# MEASURING INFORMATION IN TEXT EXPLANATIONS

## ABSTRACT

Text-based explanation is a particularly promising approach in explainable AI, but the evaluation of text explanations is method-dependent. We argue that placing the explanations on an information-theoretic framework could unify the evaluations of two popular text explanation methods: rationale and natural language explanations (NLE). This framework considers the post-hoc text pipeline as a series of communication channels, which we refer to as "explanation channels". We quantify the information flow through these channels, thereby facilitating the assessment of explanation characteristics. We set up tools for quantifying two information scores: relevance and informativeness. We illustrate what our proposed information scores measure by comparing them against some traditional evaluation metrics. Our information-theoretic scores reveal some unique observations about the underlying mechanisms of two representative text explanations. For example, the NLEs trade-off slightly between transmitting the input-related information and the target-related information, whereas the rationales do not exhibit such a trade-off mechanism. Our work contributes to the ongoing efforts in establishing rigorous and standardized evaluation criteria in the rapidly evolving field of explainable AI.

## 1 INTRODUCTION

As deep neural network (DNN) systems show superior performance on a wide variety of tasks, the explainability of DNNs has attracted increasing attention. The explainable AI (XAI) literature provides abundant methods for improving the transparency of a DNN. Among the methods that produce explanations about the decision mechanisms, text-based explanation appears particularly interesting due to its flexibility.

Text explanations mostly appear in two forms: rationale and NLE. A rationale is a subset of the input text, and an NLE is an explanation in natural language that describes the rationales (i.e., "free-text rationales"). Figure 1 shows an example.

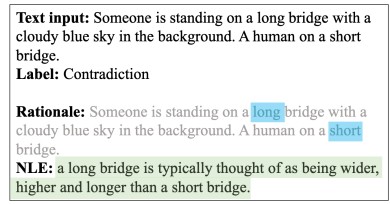

**Text input:** Someone is standing on a long bridge with a cloudy blue sky in the background. A human on a short bridge.
**Label:** Contradiction

**Rationale:** Someone is standing on a long bridge with a cloudy blue sky in the background. A human on a short bridge.
**NLE:** a long bridge is typically thought of as being wider, higher and longer than a short bridge.

Figure 1: An example of rationale and natural language explanation (NLE).

The evaluation criteria of rationale and NLE have been proposed from different routes. The approaches to evaluate the rationales include computing token-level statistics or computing the change in model performance when masking the rationales (DeYoung et al., 2020; Carton et al., 2020). Those about NLE include simulating using a proxy model and computing the utilities (Hase et al., 2020; Wiegreffe et al., 2021), computing the performance gain of student models (Pruthi et al., 2022) or computing the informativeness relative to baseline rationales (Chen et al., 2022).

We argue that the evaluations of rationale and NLE can be placed on a common ground since both text explanation approaches involve communicating the decision rationales to the readers. We abstract the two text explanation methods within a single framework based on information theory. This framework, which we call *explanation channels*, consists of three random variables: the input, the label (of the problem to be explained), and the explanan (the product of the explanation procedure, following the terminology of Hempel and Oppenheim (1948)). The explanation channels framework allows us to formulate two terms based on information theory:

- Input-explanan mutual information, which describes the *relevance* of the explanation.

- Target-explanan mutual information, which describes the explanation's *informativeness*.

These terms are deceptively hard to quantify because the input and the explanan random variables are rooted in complex distributed defined by high-dimensional data. While information theory and machine learning literature provide many tools to estimate similar terms, it is still unknown whether these tools can be used to estimate these information scores. We make it possible to estimate these MI terms. We examine the suitability of a battery of methods for this purpose and find the two most appropriate methods: InfoNCE (Oord et al., 2018) and $\mathcal{V}$-information (Xu et al., 2020).

We illustrate the validity of the MI terms with a collection of "silver labels" that are commonly used in NLP. We find that the estimated input-explanan mutual information correlates to traditional evaluation scores that measure explanations' lexical and semantic relevance. On the other hand, the estimated target-explanan mutual information describes more than just the reasoning characteristics of the explanans.

The information scores provide novel insights into the mechanisms of the explanation methods. NLEs trade-off slightly between carrying the input-related information and the target-related information, whereas the rationale explanations do not exhibit such a trade-off mechanism. Furthermore, the two MI scores reveal idiosyncratic patterns of several of the most popular contextualized language models.

In summary, we propose *explanation channels*, a framework that provides a common ground to evaluate two text-based post-hoc explanations: rationale and NLE. Our communication channel framework uncovers unique findings and contributes to the rigorous study of explanation quality, an emerging research direction that deserves more attention.

## 2 RELATED WORK

**Unified views for explanation methods**   Lundberg and Lee (2017) proposed a unified framework for several additive feature attribution methods. Ancona et al. (2018) proposed one for gradient-based feature attribution methods. Liu et al. (2021) used synthetic datasets to benchmark XAI methods and Agarwal et al. (2022) set up a public leaderboard evaluating 22 metrics. Each of those projects focused on explaining feature-based prediction systems, whereas we focus on text-based prediction systems, which do not have nominal features.

Han et al. (2022) proposed a local function approximation perspective to describe post-hoc explanation methods in a unified view, leading to a "no-free-lunch" argument for explanation methods: a locally faithful explanation may not be faithful for a distinct data distribution. Similarly, Bilodeau et al. (2022) proposed "Impossibility Theorems", stating that linear explanations may not be sufficient. We consider text-based explanations that are hard to include in unified frameworks due to the flexibility and high-dimensional nature of language.

**Information theory in NLP and XAI**   Approaches derived from information theory have been widely used in NLP. For example, *surprisal*, the negative log-likelihood of a new item following a sequence, has been used to train auto-regressive models (Radford et al., 2019). Surprisal is used to analyze patterns of texts (Meister et al., 2021) and the patterns of humans reading articles sequentially (Meister et al., 2022). Metrics derived from entropy can be used to select examples to construct prompts that maximize informativeness (Lu et al., 2022). Along these lines, we also derive scores following information-theoretic motivations.

Information theory is useful in XAI. For example, mutual information and minimum description length are used to study the informativeness of (i.e., "probe") DNN representations about some diagnostic targets (Pimentel et al., 2020; Hou and Sachan, 2021; Voita and Titov, 2020). Conditional mutual information is used to model the effects of explanation for users with different knowledge backgrounds (Jung and Nardelli, 2020).

The closest work to our paper is perhaps REV (Chen et al., 2022), which estimates the target-explanan $\mathcal{V}$-information in free-text rationales (i.e., NLEs) relative to vacuous rationales. We consider the evaluation problem from a communication channel perspective, and we measure information terms relative to null inputs (here random Gaussian vectors). Our framework additionally computes the input-explanan information, and can apply to text highlights (we refer to them as "rationales" in

this paper). Treviso and Martins (2020) formulated explanation as a sparse communication problem, where the explainer transmits information to the audience. Our framework, in contrast, considers post-hoc explanations, where the explainer is independent of the prediction model.

## 3 AN INFORMATION-THEORETIC VIEW OF XAI

### 3.1 PRELIMINARIES FOR COMMUNICATION CHANNELS

The communication channel is ubiquitous wherever information is transmitted from a source to a target. A signal is encoded at the source, transmitted through the channel, and decoded at the target. During the transmission, external signals might pollute the channel, making it a noisy channel.

Let $\mathbf{S} \in \mathbb{R}^{d_s}$ be the source and $\mathbf{T} \in \mathbb{R}^{d_t}$ be the target. When the source variable is observed, the uncertainty of the target variable is reduced. The reduction in uncertainty is the *mutual information* between the two variables, $I(\mathbf{S}\,;\,\mathbf{T}) = H(\mathbf{T}) - H(\mathbf{T}\,|\,\mathbf{S})$, where $H(\mathbf{T})$ is the entropy (uncertainty) and $H(\mathbf{T}\,|\,\mathbf{S})$ is the conditional entropy. The mutual information characterizes this communication channel's *informativeness*.

The mutual information of the communication channel is symmetric: $I(\mathbf{S}\,;\,\mathbf{T}) = I(\mathbf{T}\,;\,\mathbf{S})$. The reduction of uncertainty in $\mathbf{T}$ by knowing $\mathbf{S}$ exactly equals the reduction of uncertainty in $\mathbf{S}$ by knowing $\mathbf{T}$. Communication channels have many properties, including data processing inequality (Cover et al., 1991).

### 3.2 AN ABSTRACTION FOR THE XAI PROCESS

Here, we describe an abstraction for the procedure to explain an AI model. The model $f$ is a prediction machine that takes in the input $\mathbf{X} \in \mathbb{R}^{d_x}$ and predicts the target $Y \in \mathbb{R}$. To understand the prediction procedures, the XAI literature has proposed various methods. Each method computes an artifact, *explanan*, that explains the AI model.

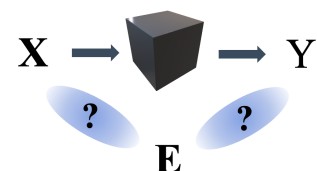

Figure 2: An illustration of the explanation channels.

A popular example trains a linear model $g$, which serves as a proxy for $f$ (Ribeiro et al., 2016). Here, the explanan is the linear model $g$. Another method, *rationale*, selects a subset of the most relevant inputs to the prediction target (DeYoung et al., 2020). Here, the explanan is one or more subsets of the input. NLE, in contrast, appears to be more flexible. Large language models (LLMs) like GPT-4 can be prompted as explainer models to generate texts with many attributes on par with human-written explanations (Wiegreffe et al., 2022). Here, the explanan is the generated text.

As shown in Figure 2, $f : \mathbf{X} \to Y$ is the "black-box" model to be explained (i.e., the *explanandum*), and $\mathbf{E}$ is the explanan. Usually, $\mathbf{E}$ is referred to as the explanation, but the term "explanation" is also used to refer to the process (Achinstein, 1983). To avoid overloading the terminologies, we refer to the product, $\mathbf{E}$, as the "explanan" throughout this paper and reserve "explanation" for the process.

Without loss of generality, we consider the explanan to be a fixed-dimensional variable: $\mathbf{E} \in \mathbb{R}^{d_e}$. In scenarios where explanans take other forms (e.g., text), one can always embed them into fixed-dimensional vectors.

### 3.3 EXPLANATION CHANNELS

The explanation of a decision-making system constitutes multiple communication channels. Two of them transmit bits of information about the system – one from the input $\mathbf{X}$ and the other from the target $Y$ – to the explanan $\mathbf{E}$. The information transmitted through the two communication channels can describe quantities that are widely concerned by the stakeholders of the decision-making system.

**Relevance** The input-explanan mutual information $I(\mathbf{X}\,;\,\mathbf{E})$ quantifies the amount of information transmitted from the input to the explanan. Given the input $\mathbf{X}$, a larger $I(\mathbf{X}\,;\,\mathbf{E})$ indicates a larger

reduction of uncertainty in the explanan. This is correlated to reduced hallucination in explanation, so we term $I(\mathbf{X} \,;\, \mathbf{E})$ the *relevance* of this explanation.

**Predictive informativeness**   The target-explanan mutual information $I(Y \mid \mathbf{E})$ quantifies the amount of information about the result of the model to be explained. A higher $I(Y \mid \mathbf{E})$ indicates that the explanan removes more uncertainty about the prediction target $Y$. A higher $I(Y \mid \mathbf{E})$ indicates that the explanation is more *informative*.

## 3.4   ESTIMATING THE RELEVANCE SCORE $I(\mathbf{X} \,;\, \mathbf{E})$

$I(\mathbf{X} \,;\, \mathbf{E})$ involves modeling two high-dimensional random variables. One method particularly suitable for such an estimation is InfoNCE (Oord et al., 2018). Given a batch of $N$ samples $\{\mathbf{x}_i, \mathbf{e}_i\}_{i=1}^{N}$, the InfoNCE estimation is:

$$\hat{I}(\mathbf{X} \,;\, \mathbf{E}) = \log N - \mathcal{L}_N, \tag{1}$$

where $\mathcal{L}_N$ is the cross-entropy loss for picking the correct $\mathbf{e}_i$ among the batch, for each $\mathbf{x}_i$:

$$\mathcal{L}_N = \frac{1}{N} \sum_{i=1}^{N} \log \frac{g(\mathbf{x}_i, \mathbf{e}_i)}{\sum_{\mathbf{x}_j \in \mathcal{X}} g(\mathbf{x}_j, \mathbf{e}_i)} \tag{2}$$

Equation 2 implicitly defines a point-wise estimation for InfoNCE, which we apply in this paper. As elsewhere (Oord et al., 2018), $g$ is a log-bilinear model parameterized by trainable parameters $W$:

$$g(\mathbf{x}, \mathbf{e}) = \exp\left(\mathbf{x}^T W \mathbf{e}\right) \tag{3}$$

Taking the average estimate $\hat{I}(\mathbf{X} \,;\, \mathbf{E})$ across all batches yields the InfoNCE estimation of the dataset. The InfoNCE estimation is a lower bound for mutual information. As the batch size $N$ increases, the lower bound becomes tighter. Please refer to Oord et al. (2018) for derivations.

Note that many alternative estimators, both parameteric (Poole et al., 2019; Cheng et al., 2020; McAllester and Stratos, 2020; Song and Ermon, 2020; Belghazi et al., 2018; Nguyen et al., 2010; Pichler et al., 2022) and nonparametric (Kandasamy et al., 2015; Kraskov et al., 2004), can estimate mutual information. On complex data distributions, parametric information estimators are usually more accurate than nonparametric ones, and this advantage improves as the data dimension further increases. Among ablation experiments on these parametric estimators, InfoNCE shows lower variances than the alternatives. We elaborate further in Appendix A.2, and defer to the recent work of Czyż et al. (2023) as a more comprehensive evaluation.

## 3.5   ESTIMATING THE PREDICTIVE INFORMATIVENESS SCORE $I(Y \,;\, \mathbf{E})$

The estimation of $I(Y \,;\, \mathbf{E})$ involves modeling a scalar random variable and a high-dimensional one. Compared to $I(\mathbf{X} \,;\, \mathbf{E})$, this scenario is more suitably estimated with another tool: predictive $\mathcal{V}$-information (Xu et al., 2020). Let $\mathbf{E}$ and $Y$ denote random variables with sample spaces $\mathcal{E}, \mathcal{Y}$, respectively. Let $\varnothing$ denote a null input without information about $Y$. Given a predictive family $\mathcal{V} \subseteq \Omega = \{h \,:\, \mathcal{E} \cup \varnothing\}$, the predictive $\mathcal{V}$-entropy is:

$$H_{\mathcal{V}}(Y) = \inf_{h \in \mathcal{V}} \mathbb{E}[-\log h[\varnothing](Y)], \tag{4}$$

and the conditional $\mathcal{V}$-entropy is:

$$H_{\mathcal{V}}(Y \mid \mathbb{E}) = \inf_{h \in \mathcal{V}} \mathbb{E}[-\log h[\mathbf{E}](Y)] \tag{5}$$

The goals of the two infimum operations are to find the predictor $h \in \mathcal{V}$ that maximizes the log-likelihood of the label data with (Eq. 4) and without (Eq. 5) the explanan $\mathbf{E}$, respectively.

We use the natural logarithm (base $e$) throughout this paper. We consider $\mathbf{E} \in \mathbb{R}^{d_e}$, and the null input $\varnothing$ to be a $d_e$-dimensional vector drawn from a Gaussian noise $\varnothing \sim \mathcal{N}(0, 0.01)$.

The predictive $\mathcal{V}$-information is defined as:

$$I_{\mathcal{V}}(\mathbf{E} \to Y) = H_{\mathcal{V}}(Y) - H_{\mathcal{V}}(Y \mid \mathbf{E}) \tag{6}$$

Similar to InfoNCE, the predictive $\mathcal{V}$-information allows a point-wise estimation. Please refer to Ethayarajh et al. (2022) for the details. The predictive $\mathcal{V}$-information is neither a lower bound nor an upper bound for the mutual information, and $I_\mathcal{V}(\mathbf{E} \to Y)$ approximates $I(Y\,;\,\mathbf{E})$ more precisely when the predictors $h$ are more high-performing (Pimentel et al., 2020; Pimentel and Cotterell, 2021).

The $\mathcal{V}$-information (also termed Bayesian mutual information (Pimentel and Cotterell, 2021) and task-specific information (Zhu et al., 2021)) has been used to study the difficulty of datasets (Ethayarajh et al., 2022), describe properties of free-text rationales (Chen et al., 2022), and characterize the informativeness of neural network representations (Pimentel et al., 2020; Hewitt et al., 2021).

## 4   DATA AND MATERIALS

### 4.1   DATA

We use the e-SNLI dataset (Camburu et al., 2018) in the ERASER benchmark (DeYoung et al., 2020). This dataset augments the SNLI natural language inference task (Bowman et al., 2015). Each instance of the language inference task presents two sentences, the premise $S_1$ and the hypothesis $S_2$, with one label $L$ describing the inference relations between them. $L$ is one of "contradiction", "entailment", and "neutral". The e-SNLI dataset covers a broad range of topics and has been a challenging evaluation for machine understanding of languages.

### 4.2   EXPLANANS

**Rationale**   The human-annotated rationales of the ERASER benchmark (DeYoung et al., 2020) specify the tokens important for decisions, while the remaining tokens are replaced with spaces.

**NLE**   We prompt ChatGPT (`gpt-3.5-turbo`) configured with the default generation hyperparameters to generate NLEs using the template:

$$\{S_1\}\{S_2\} \text{ The label is } \{L\} \text{ because} \tag{7}$$

### 4.3   SILVER LABELS FOR EVALUATING THE EXPLANS

We compute a collection of "silver labels" that describe a diverse collection of aspects of texts.[1]

**Lexical-semantic scores**   The lexical-semantic scores don't specifically evaluate the qualities of the explanations.

- **Type overlap ratio**, the portion of the word types in the text input ($S_1$ and $S_2$ concatenated) that are present in the explanan $E$. Type overlap ratio quantifies the lexical overlapping, a heuristic that many neural network NLP models rely on when learning representations (McCoy et al., 2019).
- **Edit distance ratio**, the minimum number of steps to edit the text input to acquire $E$, normalized by the text input length. This number simulates the effort in producing the explanation.
- **Cosine similarity**, the cosine similarity between the embedded input and the embedded explanan. This quantifies how the explanan is semantically similar to the explanandum.

**GPTScore labels**   Recent papers show that LLMs can evaluate text properties resembling human annotators (Zhang et al., 2020; Fu et al., 2023). We specify nine aspects in three categories: **reasoning** (informational support, causal support, convincingness, coherence), **clarity** (clarity for student, clarity for graduate), and **relevance** (label relevance, input relevance, importance), by stating each aspect with a sentence (as listed in Table 1). We use the following template to prompt the "evaluator" LLM

---

[1]Many other scores have been used to evaluate the quality of either the rationale or the NLE. The token-level F1/precision/recall scores (DeYoung et al., 2020) are suitable for rationale but not for NLE, since NLE contains too much flexibility. Additionally, these are aggregate scores, but we only consider instance-level scores.

| Category | Item | Statement |
|---|---|---|
| Reasoning | `info_support` `causal_support` `convincingness` `coherence` | The explanation provides sufficient information to support how the two sentences are associated to the label. The explanation explains why these two sentences are associated to the label. The explanation is persuasive and convinces me to believe that the question is associated to the label. The explanation bridges the gap between the two sentences and the label in a coherent and unsurprising manner. |
| Clarity | `clarity4student` `clarity4graduate` | The explanation is easy to understand for a high school student. The explanation is easy to understand for a university graduate. |
| Relevance | `label_relevance` `input_relevance` `importance` | Given the two sentences and the label, the explanation is relevant. Given the two sentences, the explanation is relevant. Ths explanation highlights the most important parts in the two sentences that associate to the label. |

Table 1: Statements describing the GPTScore evaluation items.

to score the explanan $E$ regarding a statement $A$:

> Following are two sentences, a label and an explanation.
>
> The two sentences are: $\{S_1\}\{S_2\}$
>
> The label is: $\{L\}$
>
> The explanation is $\{E\}$           (8)
>
> Please use one of 'strongly disagree', 'somewhat disagree', 'somewhat agree' and
>
> 'strongly agree' to describe your attitude towards the following statement:$\{A\}$
>
> Do not add additional words.

The model we use is InstructGPT `text-davinci-003` (Ouyang et al., 2022), which currently[2] stands in the top place at the knowledge and reasoning categories of the HELM leaderboard (Liang et al., 2022). Compared to its predecessors, `text-davinci-003` benefits from RLHF and is better at following the instructions in natural languages. It is able to follow the instructions to choose among the provided choices: only around 1 out of every 1000 results require postprocessing (e.g., stripping some additional parentheses or newline characters). Empirically, we find that the addendum to the prompt, "Do not add additional words" is helpful for encouraging it to follow the instruction.

After collecting the GPTScore labels, we map them from a categorical scale to a numerical scale. Namely: $-2$ for 'strongly disagree', $-1$ for 'somewhat disagree', 1 for 'somewhat agree', and 2 for 'strongly agree'. An exploratory analysis for inter-score correlation is included in Appendix A.5.

GPTScore is related to simulatability. Simulatability uses either humans or an LM as a proxy to predict the label from both the input and rationale (explanation), and derive criteria to measure the qualities of explanations. Simulatability measures the correlations between the rationale and the label (Chan et al., 2022). Some scores in this category include LAS (Hase et al., 2020) and its variant, RQ (Wiegreffe et al., 2021). Despite increasingly frequent anthropomorphizing claims about the LLM's capabilities, the LLMs still have significant limitations in their reasoning and explanation abilities (Dziri et al., 2023). Therefore, GPTScore labels or other scores likewise computed by LLM proxies should not be considered ground truths ("gold labels").

## 5 EXPERIMENTS

### 5.1 WHAT ASPECTS ARE THE INFORMATION SCORES MEASURING?

To query what the information scores entail, we compute the correlation of the two information scores with each of the silver labels. The correlations are plotted on Figure 3.

Additionally, we run ANOVA, which computes the portion of variance in each of the information scores that can be explained by the silver labels. The detailed procedure and results are in Appendix A.3. The following summarizes some findings.

$I(\mathbf{X} ; \mathbf{E})$ **is largely about lexical and semantic relevance** On NLE, these lexical-semantic scores can explain 46% and 43% of the total variance in $I(\mathbf{X} ; \mathbf{E})$, for Cohere and OpenAI respectively.

---

[2]As of May 1, 2023.

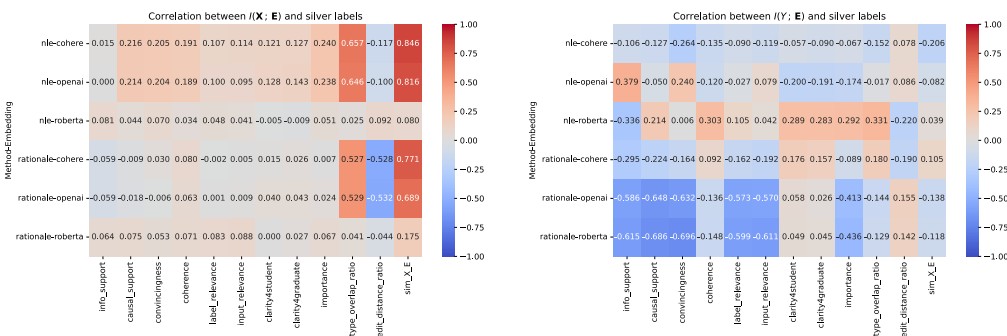

Figure 3: Correlations between relevance (left) and informativeness (right) and the silver labels.

The portions of explained variance are 31% and 17% on rationales. Other scores do not explain more than 5% of the total variance, but there is some evidence of correlations. As Figure 3 shows, the score $I(\mathbf{X}\,;\,\mathbf{E})$ shows strong correlations to the lexical and the semantic overlaps. $I(\mathbf{X}\,;\,\mathbf{E})$ positively correlates to the embedding similarity and the type overlap ratio, while negatively correlates to the edit distance ratio. On NLEs, $I(\mathbf{X}\,;\,\mathbf{E})$ show correlate mildly to the convincingness, causal support, coherence and importance scores and weak correlations to other GPTScore labels. On rationales, $I(\mathbf{X}\,;\,\mathbf{E})$ does not show correlations to the GPTScore labels. Note that the $I(\mathbf{X}\,;\,\mathbf{E})$ of the RoBERTa-embedded explanations do not show similar levels of correlations with the silver labels — we elaborate the differences between the embeddings in Section 5.3.

$I(Y\,;\,\mathbf{E})$ **is not just about the reasoning**    What the informativeness score $I(Y\,;\,\mathbf{E})$ involves varies by the explanation method and the embedding choices. The reasoning category scores can explain 16% and 21% of the variance in the estimated $I(Y\,;\,\mathbf{E})$ for OpenAI and RoBERTa on NLE (18% and 19% for rationale), and no more than 17% for any other categories. On rationales, $I(Y\,;\,\mathbf{E})$ is negatively correlated to the relevance and reasoning quality scores but is mildly correlated to the clarity scores. On NLEs, $I(Y\,;\,\mathbf{E})$ is positively correlated to the coherence, clarity, and importance scores for the RoBERTa embedding, uncorrelated for the Cohere embedding but negatively correlated for the OpenAI embedding.

## 5.2    THERE IS RELEVANCE–INFORMATIVENESS TRADEOFF FOR NLE BUT NOT RATIONALES

To further understand the phenomena described by the information scores, we compute the correlations between the relevance and the informativeness scores. The results are shown in Figure 4, and Figures 5 – 6 in Appendix.

The relevance score $I(\mathbf{X}\,;\,\mathbf{E})$ and the informativeness score $I(Y\,;\,\mathbf{E})$ show weak negative correlations for NLE. The evidence indicates that the (ChatGPT-generated) NLEs slightly trade-off between encoding the input-related information and encoding the label-related information. On rationales, the signs of such correlations differ by embeddings: negative for OpenAI, positive for Cohere, and insignificant for RoBERTa. The lack of evidence for relevance–informativeness tradeoff on rationales is likely a result of a lack of "degree of freedom", since the annotators can only select a subset of input texts to acquire the rationales.

## 5.3    ABLATION ON THE EMBEDDING METHOD

As defined in 3, we consider both the input $\mathbf{X}$ and the explanan $\mathbf{E}$ to be vectors. When the input and the explanans are both texts, there needs to be an embedding step that converts them into vectors. Embedding is the crucial component that allows multiple types of explanations to be comparable on the same ground. An embedding is already present in the $f : \mathbf{X} \to Y$ model, but sometimes this embedding is unavailable, what would be the effects if we use other embeddings to compute the information scores? We run ablation studies and query how the relevance and the informativeness scores differ.

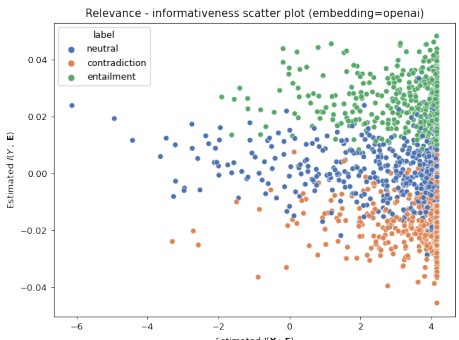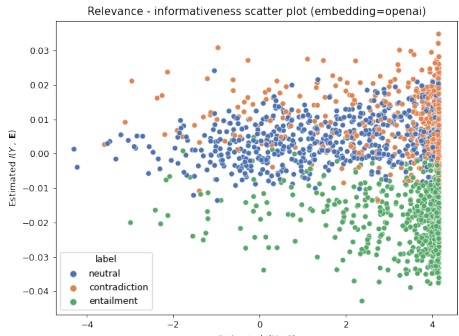

Figure 4: The relevance–informativeness scatter plots for rationale (left) and NLE (right), for the OpenAI embeddings. Spearman correlation between relevance and informativeness is $-0.0585(p = 0.0089)$ for rationale and $-0.0564(p = 0.0117)$ for NLE.

We consider three embeddings: RoBERTa (`roberta-large`) (Liu et al., 2019), OpenAI (`text-embedding-ada-002`) (Brown et al., 2020), and Cohere (`small`) (Co:here, 2023). The OpenAI embedding has $d_e = 1536$ dimensions, and the other two embeddings have $d_e = 1024$.

The Cohere and OpenAI embeddings have significantly larger relevance scores $I(\mathbf{X} \,;\, \mathbf{E})$ from the RoBERTa embedding,[3] but the difference is not significant between Cohere and OpenAI embeddings.[4] This trend holds for both the rationale and the NLE explanations.

The informativeness score $I(Y \,;\, \mathbf{E})$ score show a different pattern. For NLE, OpenAI embedding has a higher informativeness score than either Cohere or RoBERTa, which do not significantly differ.[5] For rationale, RoBERTa embedding has a significantly higher informativeness score than the other two embeddings, which do not significantly differ.[6]

We also observe that the embeddings demonstrate distinct patterns when we plot them onto a relevance–informativeness map. Figure 4 shows a relevance–informativeness scatter plot of the OpenAI embeddings. The data samples with different labels show some weak trends of clustering, but the Silhouette coefficients are weak ($-0.1088$ and $-0.0561$, for rationale and NLE, respectively). The plots of the other two embeddings are included in Figures 5 and 6 in Appendix. Cohere shows similar clustering trends as OpenAI (Silhouette coefficients $-0.0139$ and $0.0166$) embedding, but much less than the RoBERTa embedding (with Silhouette coefficients $0.1708$ and $0.7853$). A possible hypothesis to explain the inter-embedding difference is that RoBERTa strives to preserve the predictive information, embedding the texts from different classes into subspaces that are easy to separate linearly. On the other hand, OpenAI and Cohere embeddings relax this separability requirement, preserving more contextual information about the semantics.

## 6   DISCUSSIONS

**On the capacities of explanation channels**   Table 2 summarizes the relevance and the informativeness across rationale and NLE. It is perhaps surprising that the information numbers are very small, compared to the amount of information the explanation channels can potentially transmit — Two 1024-dimensional binary random variables could potentially have $1024 \times \log 2 = 618$ nats of mutual information, and the floating point variables can support an even larger channel capacity. Besides the impact of variance from the estimators, are there other factors that could explain the observations that there is so little input-explanan information $I(\mathbf{X} \,;\, \mathbf{E})$ and the target-explnan information $I(Y \,;\, \mathbf{E})$? A possible explanation is that the dimensions in the LLM's embedding vectors are highly correlated

---

[3]$p < 0.01$. All tests in this subsection are two-tailed $t$-tests. dof $= 1999$, Bonferroni corrected.

[4]$p = 0.0552$ and $p = 0.0809$ for rationale and NLE, respectively.

[5]$p = 0.0219$. After Bonferroni correction, this result is not significant.

[6]$p = 0.0674$.

| | $\hat{I}(\mathbf{X}\,;\,\mathbf{E})$ | | | $\hat{I}(Y\,;\,\mathbf{E})$ | | |
|---|---|---|---|---|---|---|
| | Cohere | OpenAI | RoBERTa | Cohere | OpenAI | RoBERTa |
| Rationale | 3.33 | 3.41 | 0.0609 | 0.208 | 0.00291 | 0.0105 |
| NLE | 2.78 | 2.88 | 0.000 | 0.0826 | $-0.00179$ | 0.0321 |

Table 2: Estimated relevance and informativeness (in nats), on the e-SNLI test set.

(an effect observed in many language models (Aghajanyan et al., 2021; Wang et al., 2020; Ethayarajh, 2019)) which reduces the overall channel capacities.

**Amount vs type of information**   Researchers have realized that the information spreading across long contexts could be "crammed" into fixed-length embedding vectors (Conneau et al., 2018). Considering the experiment findings, we could further argue that explanation does not need the full bandwidth of information of the language semantics. Providing the right *type* of information might be as important as providing the sufficient *amount*. Depending on the actual problems to explain, not all types of relevant information are appropriate — some bits of information may be disparaging, unwelcoming, and biased. The specification of, and even automatic evaluation of these attributes, however, is subject to further research. Identifying and elaborating on the types of information and their societal impacts would be crucial for understanding the explanation quality. Additionally, the explanation effect given the same amount of information describes the quality of the explanation, a term deserving more attention when developing automated approaches to explain complex decisions.

**Towards multimodal explanation channels**   Can explanation channels generalize to multimodal problems? We believe they have the potential, as long as multimodal embeddings are sufficiently informative. Recent text-to-image models like DALL-E (Ramesh et al., 2021) and image-to-text models like CLIP (Radford et al., 2021) and BLIP2 (Li et al., 2023) indicate an affirmative answer, but empirical evidence will be necessary.

## 7   CONCLUSION

We propose an information-theoretic framework, *explanation channels*, as a unified testbed for two text-based post-hoc explainable AI methods: rationale and NLE. With this framework, we can estimate the input-explanan mutual information as the "relevance score" and the target-explanan mutual information as the "informativeness score". We set up tools to compute the two scores on the explanation of natural language inference problems, which involve complex, high-dimensional distributions. By comparing to silver labels, we find that the relevance scores describe the lexical and semantic relevance scores, while the informativeness scores describe more than the reasoning qualities of the explanations. The scores reveal interesting properties of the language model embeddings and, more importantly, describe the mechanisms of multiple types of explanations. Information-theoretic frameworks have the potential to be a unified evaluation of explainable AI, empowering principled developments of trustworthy AIs.

## 8   LIMITATION

In this paper, we focus on the objective aspects and only use fully automatic evaluations to compute silver labels. Human annotators could be introduced in future studies. For the utility towards humans, we defer to the tutorial of Boyd-Graber et al. (2022). We also defer to Lombrozo (2012) for a review of the psychological aspects of explanations.

The coverage of experiments could be expanded. For example, we consider three embeddings (OpenAI, Cohere, RoBERTa-large) instead of many popular language models like LLaMA (Touvron et al., 2023) and GPT-J (Wang and Komatsuzaki, 2021). In addition to the e-SNLI, more datasets could also be experimented on. The range of text explanations can be expanded. We focus on rationale and NLE, and believe that the explanation channels framework can also be generalized to additional text-based XAI methods including contrastive explanations (Yin and Neubig, 2022), and causal explanations (Kıcıman et al., 2023).

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

# A APPENDIX

## A.1 ADDITIONAL DETAILS ABOUT THE EXPERIMENTS

**Data** We randomly sample 12k data samples, and split the dataset into 8k-2k-2k portions for train-validation-test splitting. This preserves a portion of the data that allows rigorous tests for statistical significance and keeps the various costs (mostly the costs to run API calls) reasonable.

**Runtime and computation resource** The runtimes for the MI estimators and the $\mathcal{V}$-information estimators are mostly under one minute for each pass on a T4 GPU card. The estimators running on larger batch sizes (64) take longer times to finish, but the time per pass is still under ten minutes. The short runtimes allow us to tune the hyperparameters by sweeping through a large combination.

## A.2 ADDITIONAL DETAILS ABOUT THE ESTIMATORS

**Procedure of finding the suitable estimator** For $I(\mathbf{X} ; \mathbf{E})$, we use the estimators based on the repository of Pichler et al. (2022). We select among several popular variational information estimators: CLUB (Cheng et al., 2020), DoE (McAllester and Stratos, 2020), SMILE (Song and Ermon, 2020), MINE (Belghazi et al., 2018), NWJ (Nguyen et al., 2010) and InfoNCE (Oord et al., 2018). First, we run hyperparameter tuning using Optuna (Akiba et al., 2019) through a widely applied evaluation benchmark, correlated Gaussian (Belghazi et al., 2018), and use the structural hyperparameters for each estimator based on the lowest mean squared error averaged across five scenarios.[7] These include the number of layers and whether to use layer norms and residual connections in the neural networks that parameterize each variational estimator. Then, we train the selected estimator-specific hyperparameters on the embedded e-SNLI data and select the optimal procedural hyperparameters (batch size, learning rate, and optimization epochs) based on the lowest validation loss. We refactor the scripts to output pointwise mutual information, compute on the test set and export them.

For $I(Y ; \mathbf{E})$, we implement $\mathcal{V}$-information estimators using two fully-connected predictors. The structural and procedural hyperparameters are both tuned on the validation set of the embedded e-SNLI data. The following paragraphs list more details, including the optimal configurations of the hyperparameters:

**Structural hyperparameters for variational information estimators** The search spaces for the structural hyperparameters of the information estimators are:

- CLUB: w/wo residual connections. 1-3 layers. w/wo layer norm.
- DoE: w/wo residual connections. 1-3 layers. w/wo layer norm.
- InfoNCE: without residual connections. 1-3 layers. with layer norm.
- MINE: without residual connections. 1-3 layers. w/wo layer norm.
- NWJ: choose between [GAN, JSD, X2, KL, RKL, DV, H2, W1] for the possible NWJ measures. without residual connections. 1-3 layers. w/wo layer norm.
- SMILE: choose between [None, 0.1, 1, 10] for clipping. without residual connections. 1-3 layers. w/wo layer norm.

The best structural hyperparameters are:

- CLUB: residual connection. 2 layers. layer norm.
- DoE: residual connection. 2 layers. without layer norm.
- InfoNCE: 1 layer.
- MINE: with layer norm. 2 layers.
- NWJ: NWJ measure choice W1. 1 layer. without layer norm.
- SMILE: clip choice None. 1 layer. with layer norm.

and the results are listed in Table 3.

Among the estimators, InfoNCE shows a significantly lower variance than others.

**Training-related hyperparameters** The search spaces are:

---

[7]The five scenarios are: $I = 2, 4, 6, 8, 10$, on the data dimension $d = 1024$.

| Estimator | Correlated Gaussian Minimum MSE | e-SNLI Variance on validation set | $\hat{I}(\mathbf{X}\,;\,\mathbf{E})$ on test set |
|---|---|---|---|
| CLUB | 7.82 | 820 | 119 |
| DoE | 42795.25 | 5.41e3 | -197 |
| InfoNCE | 41.08 | 0.03 | 2.06 |
| MINE | 43.88 | 1.35e3 | 2.01 |
| NWJ | 36.48 | 3.00e10 | 1.42e6 |
| SMILE | 41.53 | 3.79e5 | 105 |

Table 3: Comparisons across variational estimators with the best structural hyperparameters. On Correlated Gaussian, the MSE is the mean of five trials (against the ground truth 2,4,6,8,10). On e-SNLI, the validation set variance and the test set $I(\mathbf{X}\,;\,\mathbf{E})$ are the averages across all batches over six scenarios ({rationale,nle}×{cohere,openai,roberta}).

- Learning rate (lr): 1e-3, 3e-4, 1e-4, 3e-5, 1e-5.
- Batch size: 8, 16, 32, 64.
- Max epochs: 10.

The optimal group of training-related hyperparameter is listed in Table 4.

| Method | Embedding | Steps | Batch size | lr |
|---|---|---|---|---|
| NLE | cohere | 1250 | 64 | 1e-4 |
| NLE | openai | 1250 | 64 | 1e-4 |
| NLE | roberta | 2250 | 32 | 3e-4 |
| Rationale | cohere | 1250 | 64 | 1e-4 |
| Rationale | openai | 1125 | 64 | 3e-4 |
| Rationale | roberta | 1250 | 64 | 1e-3 |

Table 4: Optimal hyperparameters for InfoNCE.

**Optimal hyperparameters for $I(Y\,;\,\mathbf{E})$**    The search space for both the $h[\varnothing](Y)$ and $h[\mathbf{E}](Y)$ are the same:

- Batch sizes: 4, 8, 16.
- Learning rate: Between 1e-6 and 1e-2, recommended by Optuna with log scale.
- Max epochs: 10.
- Model structure: Just a linear layer that projects the d-dimensional input to the number of classes.

Table 5 lists the optimal sets of hyperparameters recommended by Optuna from 50 trials.

| Method | Embedding | $h[\varnothing](Y)$ | | $h[\mathbf{E}](Y)$ | |
|---|---|---|---|---|---|
| | | Batch size | lr | Batch size | lr |
| NLE | cohere | 16 | 0.000002 | 8 | 0.000026 |
| NLE | openai | 8 | 0.000005 | 16 | 0.000008 |
| NLE | roberta | 4 | 0.000001 | 8 | 0.000009 |
| Rationale | cohere | 8 | 0.000139 | 16 | 0.000918 |
| Rationale | openai | 16 | 0.001514 | 16 | 0.000005 |
| Rationale | roberta | 4 | 0.001761 | 8 | 0.000902 |

Table 5: The optimal hyperparameters for estimating $I(Y\,;\,\mathbf{E})$ using $\mathcal{V}$-information.

## A.3 DETAILS FOR ANOVA ANALYSIS

For each target, we run four ANOVA studies: lexical-semantic, reasoning, discourse and relevance. They are type 2 ANOVA models that can be described by the following equations, respectively:

$$\text{Target} \sim \text{type\_overlap\_ratio} + \text{edit\_distance\_ratio} + \text{cosine\_similarity}, \tag{9}$$

$$\text{Target} \sim \text{info\_support} + \text{causal\_support} + \text{convincingness} + \text{coherence}, \tag{10}$$

$$\text{Target} \sim \text{clarity4student} + \text{clarity4graduate}, \tag{11}$$

$$\text{Target} \sim \text{label\_relevance} + \text{input\_relevance} + \text{importance}, \tag{12}$$

where Target is either of $\hat{I}(\mathbf{X}\,;\mathbf{E})$ and $\hat{I}(Y\,;\mathbf{E})$. We use an off-the-shelf tool, statsmodel's `ols`, to run the ANOVA. The results are listed in Tables 6 and 7, as follows:

| Explanation | Embedding | lexical_semantics | Category reasoning | clarity | relevance |
|---|---|---|---|---|---|
| NLE | Cohere | 46 | 5.2 | 2.0 | 2.9 |
|  | OpenAI | 43 | 5.4 | 2.4 | 2.9 |
|  | RoBERTa | 2.4 | 0.42 | 0.03 | 0.07 |
| Rationale | Cohere | 31 | 1.0 | 0.2 | 0.5 |
|  | OpenAI | 17 | 0.4 | 0.04 | 0.17 |
|  | RoBERTa | 1.9 | 0.3 | 0.23 | 0.15 |

Table 6: Percentage of variance in $\hat{I}(\mathbf{X}\,;\mathbf{E})$ explained by the features in the categories.

| Explanation | Embedding | lexical_semantics | Category reasoning | clarity | relevance |
|---|---|---|---|---|---|
| NLE | Cohere | 1.9 | 3.9 | 0.66 | 0.35 |
|  | OpenAI | 3.7 | 16 | 3.8 | 7.1 |
|  | RoBERTa | 17 | 21 | 6.9 | 5.4 |
| Rationale | Cohere | 0.19 | 4.2 | 0.89 | 1.7 |
|  | OpenAI | 1.9 | 18 | 0.1 | 14 |
|  | RoBERTa | 2.6 | 19 | 1.1 | 8.4 |

Table 7: Percentage of variance in $\hat{I}(Y\,;\mathbf{E})$ explained by the features in the categories.

## A.4 RELEVANCE–INFORMATIVENESS TRADEOFF, AND THE EMBEDDING UNIQUENESS

The scatter plots of Cohere and RoBERTa are included in Figures 5 and 6.

## A.5 EXPLORATORY ANALYSIS OF THE SILVER LABEL SCORES

Table 8 summarizes some exploratory statistics for the silver label scores. Figure 7 shows the inter-score correlations for each of the silver labels.

On NLE, most scores show low inter-score correlations with each other, indicating that these scores measure a diverse collection of aspects. On the rationale explanations, however, some inter-score correlations are higher (for example, the causal support and the input/label relevance are highly correlated) because the variability of the rationale (subsets of the input texts) is lower. Regardless, the clarity and lexical overlapping scores do not correlate to most of the GPTScore results.

## A.6 EXAMPLES WITH HIGH AND LOW $I(\mathbf{X}\,;\mathbf{E})$ AND $I(Y\,;\mathbf{E})$ SCORES

Tables 9 to 12 list several examples with high and low relevance and informativeness scores (computed using Cohere embeddings), as well as some brief descriptions of the patterns of the explanations.

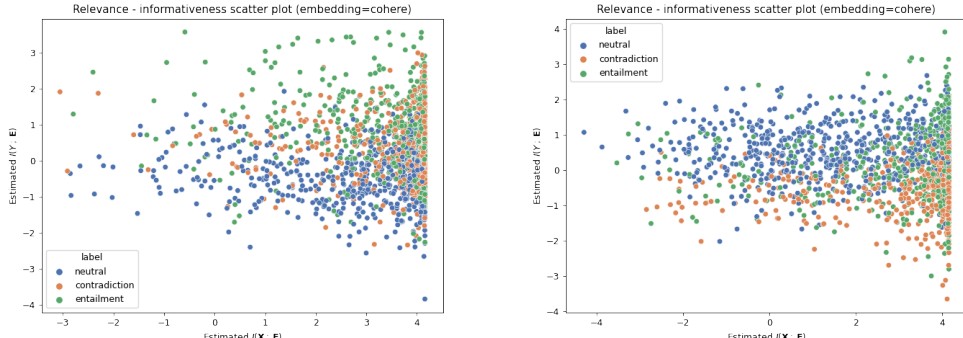

Figure 5: The relevance–informativeness scatter plots for rationale (left) and NLE (right), for Cohere embeddings. The Spearman correlations are $0.0441(p = 0.0488)$ for rationale and $-0.1416(p < 10^{-3})$ for NLE.

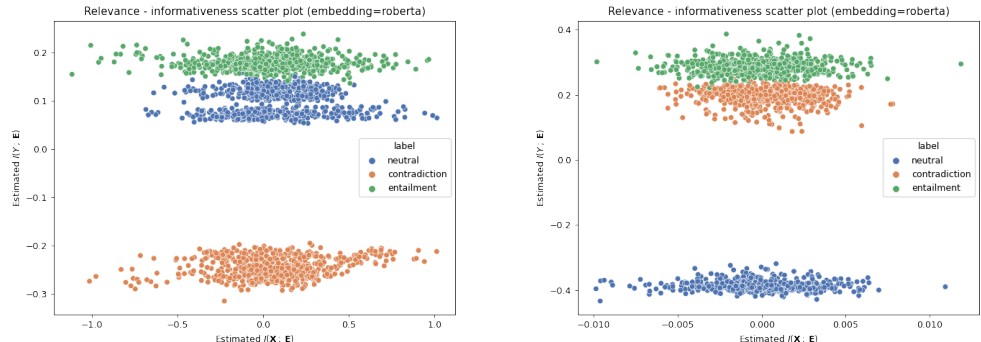

Figure 6: The relevance–informativeness scatter plots for rationale (left) and NLE (right), for RoBERTa embeddings. The Spearman correlations are $-0.0417(p = 0.0623)$ for rationale and $-0.0498(p = 0.0261)$ for NLE. Note that the Silhouette coefficients are 0.1708 (left) and 0.7853 (right), showing moderate and strong cluster effects, respectively.

| Item | Mean (Stdev) for Rationale | Mean (Stdev) for NLE |
|---|---|---|
| Edit distance ratio | 0.80 (0.14) | 1.03 (0.33) |
| Type overlap ratio | 0.22 (0.15) | 0.22 (0.18) |
| Informational support | 0.12 (1.18) | 0.57 (1.19) |
| Causal support | 0.39 (1.30) | 1.26 (0.93) |
| Convincingness | 0.11 (1.34) | 0.71 (1.40) |
| Coherence | 1.47 (0.83) | 1.65 (0.52) |
| Clarity for student | 1.34 (0.72) | 1.69 (0.48) |
| Clarity for graduates | 1.28 (0.70) | 1.62 (0.50) |
| Label relevance | 0.49 (1.35) | 1.36 (1.00) |
| Input relevance | 0.52 (1.32) | 1.39 (1.04) |
| Importance | 0.65 (1.12) | 1.20 (0.76) |

Table 8: Exploratory statistics for silver labels on the e-SNLI test sets. The GPTScore items are mapped to a numerical range between -2 ('strongly disagree') and 2 ('strongly agree').

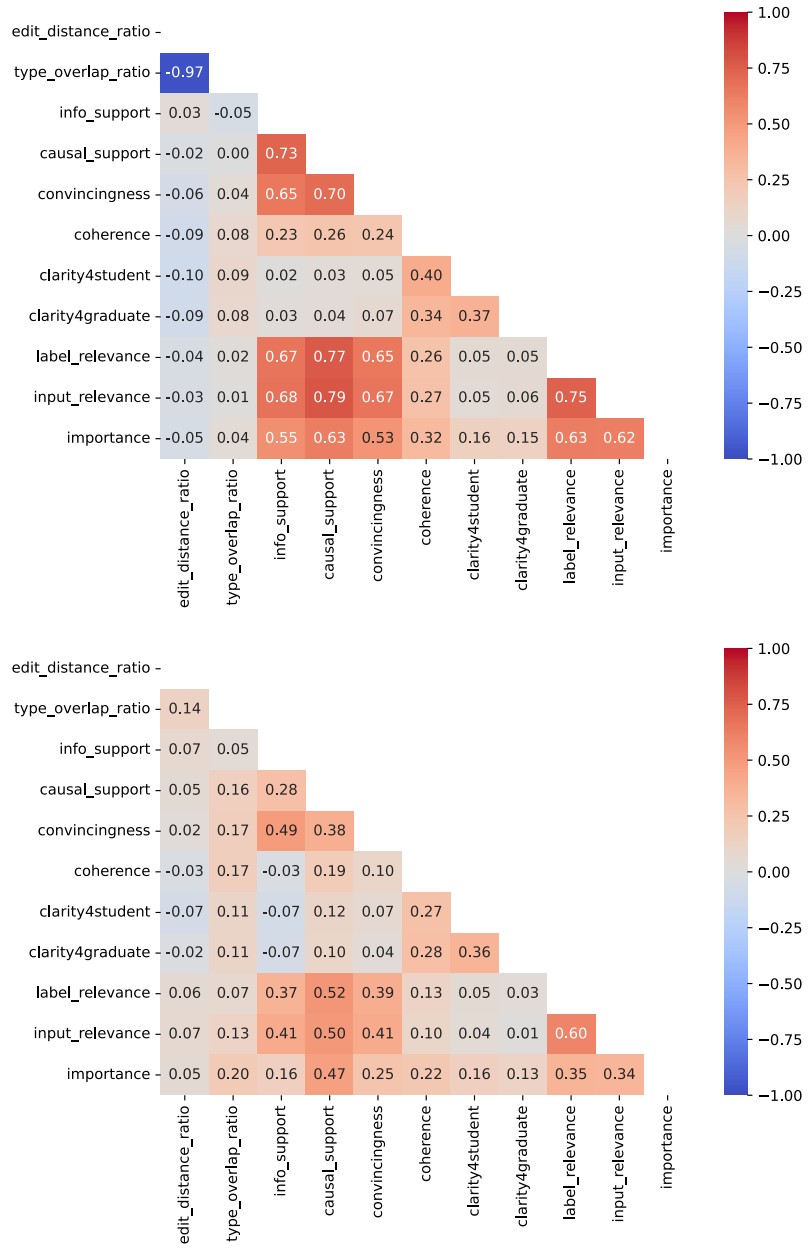

Figure 7: Inter-feature correlation heatmaps, for rationale (up) and NLE (down).

| Sentence 1 | Sentence 2 | Label | Explanation | $I(\mathbf{X}\,;\mathbf{E})$ | $I(Y\,;\mathbf{E})$ |
|---|---|---|---|---|---|
| | | | *Using NLE* | | |
| A person riding a bike on a snowy road. | A person rides a bike in winter. | entailment | the person is actively riding a bike and the weather conditions are snowy. | 4.1589 | 0.9697 |
| A man wearing a reflective vest sits on the sidewalk and holds up pamphlets with bicycles on the cover. | A man is showing pamphlets outside. | entailment | the man is showing the pamphlets to passersby, displaying that he is giving out pamphlets with bicycles on the cover. | 4.1589 | 0.7262 |
| Four women in a living room, three of which are clearly playing a musical instrument. | Four women play music together. | neutral | it does not specify what type of music or instruments the women are playing. | 4.1588 | 1.5816 |
| | | | *Using rationale* | | |
| A gymnast performing on the balance beam in front of an audience. | A person is balancing. | entailment | gymnast performing on balance beam balancing | 4.1589 | 0.8728 |
| A woman is resting against a cross shaped pole while looking towards a large body of water. | The woman is in a desert. | contradiction | woman is resting against cross shaped pole while looking towards a large body of water. in a desert. | 4.1589 | 1.7789 |
| people at beach looking for things in sand | Two people are on a sandy beach using a metal detector. | neutral | sandy beach using metal detector. | 4.1586 | 0.2028 |

Table 9: Examples of some data samples with high $I(\mathbf{X}\,;\mathbf{E})$. Usually, the NLE and rationale that receive high $I(\mathbf{X}\,;\mathbf{E})$ scores are those that attend to both the premise and the hypothesis.

| Sentence 1 | Sentence 2 | Label | Explanation | $I(\mathbf{X}\,;\mathbf{E})$ | $I(Y\,;\mathbf{E})$ |
|---|---|---|---|---|---|
| | | | *Using NLE* | | |
| An African woman with piercings, lots of necklaces and corn on her head. | A woman is carrying corn to her family. | neutral | there is no indication of any particular race, gender, or culture in the picture. | 0.0029 | 1.1132 |
| A man is bowling at an alley with cosmic imagery including bowling balls and pins above the lanes. | The man is a professional bowler. | neutral | it does not indicate any opinion about the man or the activity. | 0.0075 | 1.7032 |
| A black dog runs along the beach and jumps over a wood barrior. | a dog jumping over wood at the beach | entailment | the sentence is suggesting that it is a scene described by the sentence. | 0.0093 | 0.4059 |
| | | | *Using rationale* | | |
| The small bird is clinging to a finger of the hand holding the seeds. | A small blue bird clings to a finger. | neutral | A | 0.0993 | 0.3024 |
| An old weathered, bearded man wearing all black, riding a beautiful white donkey on white rocks. | A man is leading a donkey through a field. | contradiction | riding leading | 0.1047 | -0.5416 |
| a young boy skateboarder jumping on a platform on a skateboard. | A boy is on a skateboard. | entailment | young boy boy | 0.2127 | 1.0775 |

Table 10: Examples of some data samples with low $I(\mathbf{X}\,;\mathbf{E})$. Many NLEs receiving low $I(\mathbf{X}\,;\mathbf{E})$ scores follow a "meta-description" pattern: instead of describing the contents of the premise and the hypothesis, the NLEs describe the contexts of the contents. On the other hand, many rationales receiving low $I(\mathbf{X}\,;\mathbf{E})$ scores are succinct.

| Sentence 1 | Sentence 2 | Label | Explanation | $I(\mathbf{X}\,;\mathbf{E})$ | $I(Y\,;\mathbf{E})$ |
|---|---|---|---|---|---|
| | | | *Using NLE* | | |
| Someone sleeping on bench on city street. | That person has his eyes open | contradiction | sleeping usually requires the individual to have their eyes closed. | 3.2941 | 3.1781 |
| A small child grips onto the red ropes at the playground. | A small child eats a picnic lunch at the park. | contradiction | the image shows a child holding onto ropes but the descripton talks about the child eating. | 3.5325 | 3.1289 |
| An elderly man with facial hair and glasses, standing in his living room holding a hutch cabinet with glass doors. | The man has bought a new cabinet. | neutral | this man could fit into any age range, have any occupation and any cultural background. | 3.6478 | 2.6760 |
| | | | *Using rationale* | | |
| A blond man wearing glasses is sitting in the train looking down at something. | A blond woman wearing glasses is sitting in the train looking down at something. | contradiction | man woman | 3.4440 | 3.5600 |
| A man in a light green suit is operating machinery in a factory. | The man is outdoors. | contradiction | in a factory. outdoors. | 4.0892 | 3.5591 |
| A man in red on a bicycle rides past a glass structure. | A man is riding his bicycle. | entailment | man on a bicycle rides man is riding his bicycle. | 4.0187 | 2.9888 |

Table 11: Examples of some data samples with high $I(Y\,;\mathbf{E})$.

| Sentence 1 | Sentence 2 | Label | Explanation | $I(\mathbf{X}\,;\mathbf{E})$ | $I(Y\,;\mathbf{E})$ |
|---|---|---|---|---|---|
| | | | *Using NLE* | | |
| Black and white bird standing on hand of someone holding sunflower seeds | Two yellow birds waited on the ground for seeds to drop from a hand. | contradiction | it doesn't match the description | -2.3099 | 0.0029 |
| Little girl holding a karate trophy. | A girl won a karate tournament. | neutral | while there is likely an implication of gender, it does not explicitly state that the girl is female. | 1.5759 | 0.0093 |
| The biker is riding around a curve in the road. | The person is on a bike. | entailment | it implies that someone is riding a bicycle around a curve in the road. | 4.1466 | 0.0126 |
| | | | *Using rationales* | | |
| A dog jumps to catch a toy. | A dog is fetching a toy to earn a reward. | neutral | fetching earn a reward. | 3.8458 | 0.0009 |
| A man in an orange hard hat and vest looks at an object in his hand while the man next to him stands on a ladder in order to reach the ceiling. | Two men are working construction. | neutral | working construction | 3.9032 | 0.0037 |
| Man on four wheeler in the air. | Man is racing in the Nascar race. | contradiction | air. race. | 3.7801 | 0.0070 |

Table 12: Examples of some data samples with low $I(Y\,;\mathbf{E})$. The examples with low $I(Y\,;\mathbf{E})$ usually involve NLEs that are off-topic (the first two) or purely repeating the premise or the hypothesis (the third example). The rationales attending to only one of the two sentences are also given low $I(Y\,;\mathbf{E})$ scores, but some valid rationales (e.g., the last one) are given low $I(Y\,;\mathbf{E})$ scores too.

