# OpenReview forum: "Measuring Information in Text Explanations"
_ICLR.cc/2024/Conference — Submitted to ICLR 2024_

### Official Review · Reviewer_acAy · 2023-10-23

**Soundness:** 2 fair
**Presentation:** 2 fair
**Contribution:** 2 fair
**Rating:** 5
**Confidence:** 4

**Summary:**

This paper presents a framework for evaluating textual explanations (rationale and free-text rationale).

But I still have several questions about this paper.

1. The authors use the method of mutual information to assess the information of two text explanations, which is a common and intuitive practice. My confusion lies in how the authors estimate this mutual information. As far as I understand, using InfoNCE to estimate mutual information should require training. What is the training data? What is the training process?


2. I notice that the authors have compared several mutual information estimators in the appendix, such as club, smile, InfoNCE. Here I still have questions about the club method. According to the paper of club, the value estimated by club should be higher than the real mutual information, and thus it should be higher than the value estimated by InfoNCE, how do the authors judge that the value estimated by InfoNCE is more accurate?


3. Why use V-information instead of the traditional mutual information method?

4. Does this method depend on the accuracy of the mutual information estimator? What are its advantages over other evaluation methods (e.g., traditional metrics for evaluating NLE (rouge))?

5. In summary, I think this paper has limited innovation and usefulness.

Finally, for ICLR submissions, the appendix could be placed after the main text, not separately in the supplementary material.

**Strengths:**

See Summary for details.

**Weaknesses:**

See Summary for details.

**Questions:**

See Summary for details.

---

> ### Author Response · Authors · 2023-11-13
> **Response by authors**
>
> Thank you for reading this paper and providing comments. To the points you mentioned:
>
> - **What is the training data and procedure for InfoNCE?** We have train - validation - test splits in the dataset (Appendix A.1 elaborates more on the details). The InfoNCE, as well as other estimators, are trained on the train set.
> - **About the CLUB method, and why preferring InfoNCE over CLUB?** Appendix A.2 shows the procedure of finding one estimator among the collections. On correlated Gaussian, the mean squared errors of any estimators except DoE were acceptable. However, on e-SNLI, other estimators have much larger variance on the validation set than InfoNCE. This motivates us to choose InfoNCE rather than CLUB on the experiments (which uses e-SNLI problems).
> - **Why use V-info vs MI estimation method?** V-info has different applicability from the MI estimation problems. V-info is suitable for “vector-to-scalar” data, whereas MI estimators (CLUB, DoE, InfoNCE, MINE, NWJ, SMILE, etc.) are for “vector-to-vector” data. Sections 3.4 and 3.5 contains more formal elaborations.
> - **Does this method depend on the accuracy of MI estimator?** Yes. This method also depends on how well the embeddings encode the related information.
> - **Proposed information-theoretic scores vs other metrics?** Each metric provides a unique perspective. Our scores take root in communication channels and information theory, and reveal mechanisms about the NLEs that we consider to be quite interesting. One benefit of our score is the flexibility and the potential to generalize across multiple modalities (given suitable embedding).
> - **About the innovation and usefulness.** We consider the innovation and usefulness mainly in two aspects: (1) A step towards unifying the evaluations of two different types of text explanations. (2) Some insights about the mechanisms of the explanations, e.g., by the informativeness-relevance trade-off.

---

> > ### Comment · Reviewer_acAy · 2023-11-14
> >
> > Thanks for the reply, authors solved my problem mostly.
> >
> > But I am still confused about the question "About the CLUB method, and why preferring InfoNCE over CLUB". The authors explained that CLUB has much larger variance on the validation set than InfoNCE from the perspective of the experiment.
> > Differently, I think this may be because CLUB itself is used to estimate the upper bound of MI, while InfoNCE is used to estimate the upper bound of MI. Besides, as the batch increases, the value of MI_InfoNCE tends to log(N) ( N is the size of the batch), which may be the reason why InfoNCE has lower variance. Can the authors theoretically analyze why CLUB has much larger variance on the validation set than InfoNCE from a similar perspective?

---

> ### Author Response · Authors · 2023-11-15
> **Some analysis following this line of thought**
>
> Thanks for the recommendation. Here is an attempt following this line of thought. (Note that we do not have as strong background to carry out the theory analysis to the same level as the authors of the CLUB paper and many other papers that propose information estimator algorithms.)
>
> Many other estimators have larger variances than InfoNCE, echoing the results shown in e.g., figure 1 of the CLUB paper. McAllester and Stratos (2020) showed that lower bound MI estimation from N samples can’t be larger than O(log N), and InfoNCE is one of the lower bounds. We used batch sizes of 64 (except for NLE-roberta where we used 32), so the upper bound on the estimated amount is around a constant times 4 nats. This explains why even the most informative explanations have I(X;E) of around 4 nats (in Figures 4 and 5). Also, selecting the estimator by the least variance doesn't seem to be the optimal criterion (while it's the most convenient criterion in the absence of ground truth). Comparing the behavior of CLUB to InfoNCE, and other estimators on the explanation data has a lot of research opportunities left in the future works. We will add these discussions to the papers.

---

> > ### Comment · Reviewer_acAy · 2023-11-16
> >
> > Thanks for the response. I hope that in the revised version, the authors can combine the theory of mutual information estimation and experiments to show why preferring InfoNCE over CLUB. It will be beneficial for the reliability of the proposed metrics.
> >
> > Finally, I will improve my score.

---

### Official Review · Reviewer_7iHa · 2023-10-31

**Soundness:** 2 fair
**Presentation:** 1 poor
**Contribution:** 1 poor
**Rating:** 5
**Confidence:** 4

**Summary:**

This paper proposes to view text explanations in an information theory based framework. This paper focuses on the rationale and NLE types of text explanations for text classification (specifically the e-SNLI dataset). After formulating the investigated cases within the information theory based framework, the authors propose to measure the mutual information (1) between the input and the explanan (called relevance score) and (2) between the explanan and the target (called informativeness score), and analyze the correlation among the proposed relevance score, informativeness score, and multiple proposed silver labels. The goal of this paper, to my understanding, is to propose the relevance and informativeness scores for future explanation evaluation.

**Strengths:**

* The way of forming the text explanation within the information theory framework is interesting.
* The experiments have included three embedding models: RoBERTa, OpenAI, Cohere, to test the generalizability of the proposed relevance and informativeness scores, which highly depends on the used embeddings.
* The experiments have considered two often-seen types: the rationale, which is defined to include the tokenwise explanation, and the NLE, which is defined to be explaining the input-target relationship via natural language description.

**Weaknesses:**

* The writing can be improved: The goal of this paper can be more clear. I was lost in the middle of the paper and wondered (1) which parts are used for evaluation and (2) what are they truly evaluating for? I could only realize the goal after reading through the whole paper and gave it a guess.
* This paper claims to unify the evaluation of rationale and natural language explanations. Unification would make me expect the proposed evaluation method can evaluate them in a good standing. However, the results turn out the proposed metrics do not have a consistent meaning for human interpretation and meanwhile there is no user study to validate this.
* From the results, the most consistent part is that the proposed relevance score I(X;E) is correlated with the type overlap ratio and embedding similarity. Nonetheless, since explanan often includes (similar) words in the input, the result is not giving new insights.

**Questions:**

* Is there a reason for choosing embeddings for computing the entropy? What do the authors think about using the probability distribution to compute the entropy?
* Please add proper reference to the statistics in the paper:
  *In section 4.3: Where are the statistics for the statement “On the explanations in natural language, most inter-score correlations are low, indicating that these scores measure a diverse collection of aspects.”? The authors have pointed to Appendix A.5, but it could be better to point to Figure7. Also, I would suggest this statistics be moved to the main content.
  * In section 5.1: How do the authors compute the statistics “The reasoning category scores can explain 16% and 21% of the variance in the estimated I(Y ; E) for OpenAI and RoBERTa on NLE (18% and 19% for rationale), and no more than 15% for any other categories.”?
  * In section 5.2: The authors do not put any reference to statistics in this section. I guess the authors are referring to Figure 4, 5, and 6. I would also suggest Figures 5 and 6 be moved to the main content. To comprise the paper space, I would think that prompt templates and discussion about multimodal can be moved to appendix.
* I would suggest changing the naming of “informativeness” in the GPTScore evaluation items. This term is the same as the proposed “informativeness score I(Y;E)”, so can cause confusion.
* Why is there no “informativeness” from Table1 in Figure3’s x-axis?
* I would suggest changing the order of x-axis in Figure3 to match the order of them in Table1.

---

> ### Author Response · Authors · 2023-11-13
> **Response by authors**
>
> Thank you for considering this paper interesting, and for commending on the empirical solidity of this paper! To the points mentioned in weaknesses and questions:
>
> - **The evaluation plan and goals.** We added the experiment protocol and goals to the first paragraphs in 5.1 to 5.3. We also updated the pdf to make it more organized.
> - **Only automatic and no human evaluations?** Yes — human eval of explanation is a huge topic. Each format of explanation format (NLE or rationale), and each topic of the explanation contents can have a unique impact to the audience [1]. Also the situation of the audience matters [2]. The human evaluations in a careful enough manner would lead to the scales of experiments that significantly go beyond the scope of this paper, so we can only focus on the automatic evaluation perspective.
> - **New insights beyond the correlations to some scores?** There are at least three insights beyond correlations: (1) With ANOVA, we quantify how much each aspect (lexical-semantic, reasoning, clarity, relevance) can explain the variation in the measured information scores. (2) We show a tradeoff between relevance-informativeness phenomenon. (3) We illustrate that the openai and cohere embeddings preserve the distributions, whereas the roberta embeddings focus more on the decision boundary.
> - **Why choose embeddings to compute the entropy?** Embedding is the crucial component that allows multiple types of explanations (NLE, rationale, and potentially other types) to be comparable on the same ground. The pipelines involving conditional probabilities usually need another module that computes the one-dimensional probability value from the embeddings.
> - ******************************************************Comments about the formats:******************************************************
>     - Reference for “Inter-score correlations are low”: Indeed. They are from Figure 7. We updated the references in the pdf to make this clearer.
>     - References for the ANOVA results: We updated the references to the ANOVA details to make this clearer.
>     - Figures 4-6, and moving some statistics to the main content. We moved around some parts. Figures 5-6 takes more space than the prompt templates (which we also consider quite important, since LLM’s behavior is heavily affected by the actual wording of the prompts) so we are keeping them at the current version.
>     - Naming confusion for “informativeness” (GPTScore vs I(Y;E)). Good catch. We changed to informational support (`info_support` in the tables to save space) to avoid confusion.
>     - Table 1 in Figure 3 x-axis: The “informativeness” (now info_support) is added now. The past version didn’t contain this due to a confusion with I(Y;E).
>     - Changing the order of x-axis in Figure 3 to match the previous table: Done.
>
> References:
>
> [1] For example: Chen, Chacha, et al. "Machine explanations and human understanding." *arXiv preprint arXiv:2202.04092* (2022). The “How are explanations useful to humans?” section in this reading list compiles some papers on this topic: [https://ziningzhu.notion.site/Explanations-reading-list-56cd203b1d1c4fd79e8fcf319b1560a8?pvs=4](https://www.notion.so/Explanations-reading-list-56cd203b1d1c4fd79e8fcf319b1560a8?pvs=21)
>
> [2] Zhu, Zining, et al. "Situated Natural Language Explanations." *arXiv preprint arXiv:2308.14115* (2023).

---

> > ### Comment · Reviewer_7iHa · 2023-11-22
> > **Thanks to authors' response**
> >
> > Thanks to the authors for their responses and paper revision. Since the paper writing (one of my concerned weaknesses) has improved and most of the confusion in my original questions has been tackled, I may raise my score from 3 to 5. However, I am not sure to further raise the score since my other two concerned weaknesses haven’t been addressed.
> >
> > From my observation, a major part of the paper’s contribution is claimed to be the introduction of the relevance and informativeness scores. From the paper, the two scores are compared to the defined silver labels, which are also a type of automatic evaluation metrics. Both the relevance and informativeness scores do not have apparent correlation with certain types of silver labels, other than the type overlap ratio.
> >
> > Therefore, (1) based on the results, I’m not sure why using the two scores instead of silver labels themselves as the evaluation metrics for natural language explanations, (2) I’m not sure if the two scores can have other meanings if possibly compared to other defined human evaluations.
> >
> > From the paper and authors’ responses, I still conclude my considering strengths for this paper is their introduced framework and the diverse experiments. Nonetheless, I’m not giving a higher rating since a major part of the paper’s claim is the introduced scores, which I found are not well validated by experiments yet.

---

> ### Author Response · Authors · 2023-11-17
> **More about the meaning of the two proposed metrics**
>
> Dear Reviewer 7iHa,
>
> Following the previous response, we would like to further discuss the meaning of the two information-theoretic scores and put them into context.
>
> The two scores take root in the communication channels: one unit of information (bit) quantifies the amount of information that helps us remove the uncertainty of a fair coin toss. (Machine learning uses ln instead of log, which we follow. It's just differing by a constant.) Intuitively, there should be a score that describes how much information is "transmitted" through each channel in the communication. We apply information estimation tools and showed it's possible to do so.
>
> We call the I(X;E) score relevance, since this score describes more of how the explanan relates to the input. The experiments verify that the I(X;E) is correlated to many lexical-semantic features, especially the cosine similarity. Additionally, the features in the lexical-semantic category can explain a high portion of the variance in I(X;E), illustrating that I(X;E) describes more about the relatedness of the input.
>
> We call the I(Y;E) score predictive informativeness, because (1) the computation approach uses predictive V-info, and (2) this score describes more of how the explanan predicts the output label. To predict the label (i.e., do the task specified by the dataset), a wide range of mechanisms might happen under the hood, including reasoning, inferring, pattern matching based on the embedding spaces, etc. These are illustrated from our correlation experiments and the analysis of variance experiments. Overall, it's hard to claim that this score is about e.g., reasoning, so we do not draw such a conclusion.
>
> We hope the above elaboration clarifies a bit. If there are further questions, please do not hesitate to follow up on the comments.

---

### Official Review · Reviewer_gJuB · 2023-11-01

**Soundness:** 3 good
**Presentation:** 3 good
**Contribution:** 3 good
**Rating:** 8
**Confidence:** 4

**Summary:**

The authors propose a framework that considers the post-hoc text pipeline as a series of communication channels. They quantify the information flow through these channels and facilitate the assessment of explanation characteristics, quantifying two information scores: relevance and informativeness. They illustrated the proposed information score measures by comparing them against traditional evaluation metrics.

**Strengths:**

- the paper is original - we found only a paper with some similar concepts used in a different context
 - the authors tackle a relevant problem and propose a framework to evaluate the informativeness of text explanations. Furthermore, the framework contemplates a high degree of automation, making it feasible to deploy in production settings.
 - the authors propose measuring two key aspects of the explanations: relevance and predictive informativeness.
 - we consider the paper to be of good quality: they performed a good overview of related work, all of the claims are supported with experimental results, acknowledged limitations, and ensured the presentation is clear.

**Weaknesses:**

- the structure of how experiments and results are reported can be improved. In particular, it would be helpful if the authors list the experiments performed, listing rationale behind the experiment, the procedure, aims, metrics, and other aspects of relevance.

**Questions:**

We consider the paper interesting and relevant. Nevertheless, we would like to point to the following improvement opportunities:
Data and Experiments
   	- We encourage the authors to restructure Section 4 and Section 5 to describe better (a) the original data they have and (b) the experimental design (rationale behind the experiment, procedure, aims, metrics, etc.). In the current manuscript, it seems most of the experimental design is described in the "Data and Materials" section, while the "Experiments" section resembles more to "Results and Evaluation".
   	- Why do the authors report Spearman and not Kendall correlation? Did they check for the Spearman correlation assumptions?Figures:
  - Figure 4: the authors provide two plots with identical descriptions, but from the caption, they seem to refer to different concepts (one should reflect informativeness, while the second reflects rationale)?

---

> ### Author Response · Authors · 2023-11-13
> **Response by authors**
>
> Thank you for commending on the novelty, feasibility, and the quality of the paper! To the points in weaknesses and questions:
>
> - **Improve the structure of experiments and results.** Thanks — we updated the pdf. The current version lists the sections in a more organized way.
> - **Figure 4 name confusion**: Yes this confusion is avoided in this version. We changed the “informativeness” in the silver labels into “informational support”.

---

> > ### Comment · Reviewer_gJuB · 2023-11-23
> >
> > We authors have tackled our comments. We have reviewed the comments from other reviewers and have no further observations.

---

### Official Review · Reviewer_21zw · 2023-11-01

**Soundness:** 3 good
**Presentation:** 2 fair
**Contribution:** 4 excellent
**Rating:** 8
**Confidence:** 2

**Summary:**

This paper explores a novel, information theoretic approach for evaluating natural language explanations. Both rationale-based and natural language-based explanations are considered.  The paper speaks of an "explanan" - the "product" (as opposed to the process) that is the explanation.  The inputs and outputs of the any explanan can be reduced to a fixed dimension representation by conventional text embedding tools, to which mutual information approximations can be applied.  These are two - one input based, the "relevance"; and the other output-based, the "informativeness."

In summary by considering these two measures as information channels, the paper finds that relevance is related to traditional measures of relevance, and informativeness is related to the explanans' reasoning.

**Strengths:**

Text-based explanation is a nascent field for which this paper offers a novel attempt.   The paper shows the feasibility of applying mutual information measures, noting that currently it is "still unknown whether these tools can be used to examine information scores." The paper's demonstration of the feasibility of using various approximations to mutual information in this case is novel. The use in practice of these measures for evaluation is a worthwhile contribution. The entire field of natural language model evaluation is a developing area, unlike the mature methods used in supervised machine learning that have propelled that field forward.

**Weaknesses:**

The conclusions are modest, and give limited insight.  Despite this the approach has promise, as it plows new ground in an area where there is limited success today.

**Questions:**

It is not clear how the dataset - benchmark, the "silver labels" and the language models come together in the experiments. The experiments section does not describe the process  - how is evaluation applied?

---

> ### Author Response · Authors · 2023-11-13
> **Response by authors**
>
> Thank you for commending on the contribution! We believe that model evaluation can benefit from information-theoretic approaches. While the area of information estimator is not as popular as e.g., prompting LLMs, or LLM agents, there are still some meaningful progress in the past a few years, and we still believe there will be improvement potential.
>
> To your question about the experiment procedures: we updated the pdf. Each subsection there now starts with a paragraph describing the experiment setup. The procedure is now clearer and reflects how each experiment is set up.

---

### Author Response · Authors · 2023-11-14
**Correcting a missing term in the ANOVA equation for lexical-semantic category**

Dear reviewers,

Upon carefully checking of the details, we found that a term, type_overlap_ratio, was missing in the ANOVA equation for the lexical-semantic category (Appendix A.3 equation (9)). After correcting this typo, the interpretations drawn from these numbers remain unchanged. We are uploading a corrected pdf as a revision.

Following are the details of the correction.

Equation (9) before: `Target ~ edit_distance_ratio + cosine similarity,`

Equation (9) corrected: `Target ~ type_overlap_ratio + edit_distance_ratio + cosine_similarity,`

This correction changes the values for the lexical_semantics column in Tables 6 and 7. Following are the diffs for the column in Table 6 $\hat{I}(\mathbf{X};\mathbf{E})$:

| Explanation | Embedding | lexical_semantics (after) | lexical_semantics (before) |
| --- | --- | --- | --- |
| NLE | Cohere | 46 | 60 |
| NLE | OpenAI | 43 | 57 |
| NLE | RoBERTa | 2.4 | 2.5 |
| Rationale | Cohere | 31 | 33 |
| Rationale | OpenAI | 17 | 18 |
| Rationale | RoBERTa | 1.9 | 1.9 |

The percentage of variances explained for Cohere and OpenAI embedding in NLE are lower when the missing term is included, but these are still much larger than the variances in other categories. The conclusion drawn from these numbers (I(X;E) is largely about lexical and semantic relevance) remains unchanged.

Following are the diffs for the column in Table 7 $\hat{I}(Y;\mathbf{E})$:

| Explanation | Embedding | lexical_semantics (after) | lexical_semantics (before) |
| --- | --- | --- | --- |
| NLE | Cohere | 1.9 | 3.6 |
| NLE | OpenAI | 3.7 | 1.5 |
| NLE | RoBERTa | **17** | 0.39 |
| Rationale | Cohere | 0.19 | 0.89 |
| Rationale | OpenAI | 1.9 | 0.77 |
| Rationale | RoBERTa | 2.6 | 9.2 |

There is one significant change: NLE with the RoBERTa embedding. The lexical-semantics category score actually explains 17% of the variance, rather than 0.39%. This may result from the fact that the type_overlap_ratio (that was previously missing) has a slightly larger correlation with I(Y;E) than most other silver labels (Figure 3). The conclusion drawn from these numbers (that I(Y;E) is about more than the reasoning) remains unchanged.

We also updated the numbers in the corresponding texts in the new pdf version. We apologize for the confusion due to the missing term.

---

### Meta-Review · Area_Chair_F3KS · 2023-12-11

**Metareview:**

The paper presents an information-theoretic approach using communication channels for evaluating the relevance and informativeness of natural language explanations and rationales contained in their high-dimensional embeddings. Experiments are conducted on a single benchmark, e-SNLI.

Strengths: The paper addresses an important area of research, given how commonplace text explanations are in explaining AI model outputs, especially for LLMs.

Weaknesses: It is not clear how this work empirically compares to Chen et al.’s, which the authors recognize is a closely related approach for evaluating text explanations. In the absence of baselines, it is not clear what insight can be gained from using the proposed approach. The writing and the organization of the paper was also hard to follow. It was particularly hard for reviewers to follow the experimental design. While we recognize the merits of the authors in updating the results found in the rebuttal period, they substantially change the original results, giving the impression that further analysis might be necessary.

**Justification For Why Not Higher Score:**

See weaknesses above, the lack of empirical comparison to prior work and somewhat opaque experimental design came across as primary weaknesses.

**Justification For Why Not Lower Score:**

N/A

---

### Decision · Program_Chairs · 2024-01-16

Reject